# Fast continuous streaming sort in big streaming data environment under fixed-size single storage

**Suluk Chaikhan**, **Suphakant Phimoltares** *, **Chidchanok Lursinsap**

Advanced Virtual and Intelligent Computing (AVIC) Research Center, Department of Mathematics and Computer Science, Faculty of Science, Chulalongkorn University, Bangkok, Thailand

* suphakant.p@chula.ac.th

**Data Availability Statement:** All Delhi Weather Data files are available from the kaggle.com database (https://www.kaggle.com/mahirkukreja/delhi-weather-data). All Artificial Lunar Landscape Dataset files are available from the kaggle.com

## Abstract

Big streaming data environment concerns a complicated scenario where data to be processed continuously flow into a processing unit and certainly cause a memory overflow problem. This obstructs the adaptation of deploying all existing classic sorting algorithms because the data to be sorted must be entirely stored inside the fixed-size storage including the space in internal and external storage devices. Generally, it is always assumed that the size of each data chunk is not larger than the size of storage ($M$) but in fact the size of the entire stream ($n$) is usually much larger than $M$. In this paper, a new fast continuous streaming sorting is proposed to cope with the constraint of storage overflow. The algorithm was tested with various real data sets consisting of 10,000 to 17,000,000 numbers and different storage sizes ranging from $0.01n$ to $0.50n$. It was found that the feasible lower bound of storage size is $0.35n$ with 100% sorting accuracy. The sorting time outperforms bubble sort, quick sort, insertion sort, and merge sort when data size is greater than 1,000,000 numbers. Remarkably, the sorting time of the proposed algorithm is 1,452 times less than the sorting time of external merge sort and 28.1767 times less than the sorting time of streaming data sort. The time complexity of proposed algorithm is $O(n)$ while the space complexity is $O(M)$.

## Introduction

Sorting is one of the fundamental algorithms of various applications such as fast searching, computing quartiles, and finding duplicates of a given number set. Many previously proposed classic sorting methods, e.g. quick sort, heap sort, and merge sort, are very efficient in terms of time complexity. However, these methods are based on the strong conventional assumption that the entire set of data must be stored in a processing memory, including internal and external memory, during the sorting process. Storing the entire data set prior to the sorting process is feasible if the data size does not overflow the physical capacity of memory. But due to the present disruption of big data, it is impossible to retain the entire data set in the memory during the sorting process. Big data are continuously generated in various applications such as internet communication, sensor networks, and pattern cognition in artificial intelligence

database (https://www.kaggle.com/romainpessia/artificial-lunar-rocky-landscape-dataset). All Continue playing game files are available from the kaggle.com database (https://www.kaggle.com/qichenghu/continue-playing-game). The authors confirm that anyone can access these third-party data in the same manner. The authors confirm that they did not have any special access privilege.

**Funding:** Suluk Chaikhan received a grant from Development and Promotion of Science and Technology Talents Project (DPST). Chidchanok Lursinsap received a grant from Thailand Research Fund under grant number RTA6080013. The funders had no role in study design, data collection and analysis, decision to publish, or preparation of the manuscript.

applications. Using massive data is a major issue in various branches of the computational sciences [1–3]. As the growth of data size exponentially escalating [4–6], data computing and analysis are becoming extremely difficult, resulting in the need for huge memory resources for the sorting process and massive storage for storing the results [7, 8]. Current VLSI technological development to increase the capacity of memory and storage falls behind the explosion of data consumption demand [3, 9, 10]. The telecommunication industry, business intelligence, data mining, cloud computing, mobile technology, and machine learning are confronted with the issue of memory overflow crisis when processing huge amounts of data [11–14]. This obstruction causes a severe data loss and induces the incorrect results for solving a problem. Hence, a new sorting algorithm capable of handling memory overflow must be developed. The time complexity of this algorithm must not exceed the worst time complexity of classic sorting algorithms.

Although classic sorting algorithms such as bubble sort, insertion sort, merge sort, and quick sort are popular, they have been improved to work faster. The time complexity is always the main focus of improvement. Singh et al. [15] proposed a new algorithm, namely, RVA sorting based on bubble sort and quick sort. The number of comparisons and the execution time are less than those of the existing algorithms. Wild et al. [16] analyzed the speed of Yaroslavsky's quick sort in terms of the number of comparisons, swaps, and bytecodes. The cost of Yaroslavsky's algorithm can be converged by a fixed-point equation. Agrawal and Sriram [17] proposed a new technique called the Concom sorting algorithm based on a comparison and counting method. The results of the execution time and comparison are better than bubble sort and insertion sort. Osama et al. [18] proposed a mapping sorting algorithm based on mapping technique. The mapping is not related to comparison or swapping, but it approximates temporary order by a linear equation. The wrong values at any positions are corrected through swapping or rearranging steps. Vignesh and Pradhan [19] improved merge sort by using multiple pivots to sort data in the array. The time complexities of the best and worst cases are $O(n)$ and $O(n \log n)$, respectively. Idrizi et al. [20] modified the counting sort algorithm by reducing the number of comparisons and the execution time. The sorting result surpasses quick sort, bubble sort, and merge sort. Mohammed et al. [21] proposed bidirectional conditional insertion sort that improves insertion sort and reduces the number of comparisons by using two pivots as left-comparator and right-comparator. Goel and Kumar [22] proposed two new sorting algorithms called Brownian motus insertion sort and Clustered binary insertion sort to reduce the number of comparisons. The time complexity of first algorithms is $O(\sqrt[0.54]{n})$ while the second algorithm takes $O(n \log n)$. Omar et al. [23] generated a new algorithm called double hashing sort based on mapping sort algorithm and hashing technique. The proposed algorithm can reduce both the number of comparisons and the execution time. Recently, new sorting algorithms have been proposed to handle big data with small primary memory size by using new auxiliary space as secondary memory or hard disks to merge all sorted parts of big data. Zushi and Goswami [24] proposed a new quadratic sorting algorithm by considering an unsorted sequence as a set of disjoint sorted sequences. The execution time of the proposed algorithm is more efficient than those of insertion sort, bubble sort, selection sort, and quick sort. Gugale [25] proposed super-sort sorting algorithm by picking sorted sequence in unsorted sequence. This concept can reduce the number of steps in sorting process. The time complexity of worst case is $O(n \log n)$. Lee et al. [26] proposed ActiveSort for external sorting using SSDs. The concept involves using ActiveSort to reduce the number of read/write operations. This can reduce the amount of write operations to 40.4% of the original value. Laga et al. [27] proposed MONTRES, external sorting algorithm on SSDs to reduce the run time of sorting process by decreasing read and write costs of input and output operations and the size of

temporary data. Liang et al. [28] designed B*-sort, a new sorting algorithm, which can reduce the amount of write traffic on Nonvolatile random access memory (NVRAM). Moreover, the structure used in this algorithm is similar to the binary search tree, which takes the time complexity of $O(n \log n)$. Arge and Thorup [29] studied I/O and RAM models to design I/O-efficient as well as RAM-efficient priority queues, and they also analyzed the lower bounds of the models. Kanza and Yaari [30] proposed a new external sorting method based on flash storage that avoids writing to the hard disk. In addition, Elder and Goh [31] designed a new sorting algorithm by a finite stack and an infinite stack on permutation sequence. Although this algorithm can sort streaming data, the sorting result may be not correct in terms of order and value.

It is noticeable that all classic sorting algorithms are designed by storing the entire data set prior to the sorting process. This implies that the data size must be at most as large as the memory size. On the contrary, streaming data continuously enter into the memory as a chunk whose size cannot exceed the memory size at any time. Obviously, any classic sorting algorithm cannot achieve the accurate sort because the algorithm sorts only the data presently stored in the memory. The rest of data not entering the memory are not sorted. One potential solution to resolve the problem of classic sorting algorithms is to use external sorting. However, all external sorting algorithm must rely on the size of external storage. With current data storage technology, the increase of storage size cannot keep pace with the increase of generated data, such as internet data [32, 33].

Recently, Chaikhan et al. [34] proposed an algorithm called *streaming data sort* to continuously sort incoming big streaming data by using a uniprocessor and only one internal fixed-size working memory. The memory size is much smaller than the total streaming data size. No external storage for storing overflowed data is required. The data flow into the working memory as a streaming chunk. After sorting the incoming data, these data are discarded and represented in forms of compact and single groups. The sorting time complexity is $O(n)$, where $n$ is the size of streaming data and the space complexity is $O(M)$, where $M$ is the size of working memory. Moreover, unlike the external sort, *streaming data sort* does not require any external space for the merge process. *Streaming data sort* merges sorting results and current incoming data based on a limited memory that exists at the beginning of the process, whereas the merging space, which is always much greater than the size of huge data, is strongly required for the external sort. Although *streaming data sort* can efficiently solve the sorting problem with limited memory size, the algorithm is limited to some specific distances between any two consecutive numbers and remains incapable of handling duplicate numbers. Let $n$ be the size of streaming data, $M \ll n$ be the size of working memory, $k \geq 1$, and $\epsilon > 0$. The comparison of existing sorting algorithms and our *fast streaming data sort* are shown in Table 1, which contains eight columns: (1) compatibility with duplicate data, (2) stability, (3) compatibility with streaming data, (4) extra storage needed, (5) working space, (6) time complexity, (7) correct orders, and (8) correct values.

Data lifecycle [35–39] is another essential issue in data analysis and data mining. Each datum passes various stages during its lifetime, starting from creation, analysis, and so on until expiration, possibly due to being obsolete. In practice, any expired data must be removed from the relevant database to gain available space and to keep the data up-to-date. Streaming data also possess this lifecycle trait without exception. The method of *streaming data sort* [34] is incapable of managing expired data after being sorted and captured in a compact representation. However, this issue is resolved in this study.

The rest of this paper is organized into the following sections. The "Studied Problems and Constraints" section discusses the problem studied in this paper and the concerned constraints. The "Background" section summarizes the related works. The "Definitions and

**Table 1. Comparison of sorting algorithms on streaming data.**

| Sorting algorithms | Compatibility with duplicate data | Stability | Compatibility with streaming data | Extra storage needed | Working space | Time complexity | Correct orders | Correct values |
|---|---|---|---|---|---|---|---|---|
| Bubble sort | Yes | Yes | No | No | $n + \epsilon$ | $O(n^2)$ | Yes | Yes |
| Selection sort | Yes | No | No | No | $n + \epsilon$ | $O(n^2)$ | Yes | Yes |
| Insertion sort | Yes | Yes | Yes | No | $n + \epsilon$ | $O(n^2)$ | Yes | Yes |
| Quick sort | Yes | No | No | No | $n + \epsilon$ | $O(n^2)$ | Yes | Yes |
| Merge sort | Yes | Yes | No | Yes | $n + \epsilon$ | $O(n \lg n)$ | Yes | Yes |
| Heap sort | Yes | No | No | No | $n + \epsilon$ | $O(n \lg n)$ | Yes | Yes |
| External sorting | Yes | Yes | Yes | Yes | $kn + \epsilon$ | N/A | Yes | Yes |
| Permutation sort [31] | Yes | Yes | Yes | Yes | $kn + \epsilon$ | N/A | No | No |
| Streaming data sort [34] | No | Yes | Yes | No | $\leq 0.35n$ | $O(n)$ | Yes | Yes |
| **Fast streaming data sort** | Yes | Yes | Yes | No | $\leq 0.35n$ | $O(n)$ | Yes | Yes |

$n$ is the total streaming numbers to be sorted, $M \ll n$ is the limited size of working storage, $k \geq 0$, and $\epsilon > 0$.

Notations" section defines the notations used in this paper. The "Concepts" section explains the concepts of the proposed approach. The "Proposed *fast streaming data sort* Algorithms" section provides the details of proposed algorithms. The "Experimental Results and Discussion" section provides the experimental results for the proposed algorithms as well as the discussion of the results. The "Conclusion" section concludes the paper.

## Studied problems and constraints

The basic studied problem in this paper focuses on the sorting of streaming data under the constraints of limited working memory size and a uniprocessor similar to the problem studied in [17]. Although the sorting algorithm in [34] can efficiently sort the streaming data under the imposed constraints, it assumes that there are no duplicate data in the streaming input. This assumption is impractical for many real-world data sets. Furthermore, the condition of data lifecycle was not taken into account. On the contrary, this study concerns the issues of duplicate data and data lifecycle as new constraints. Only integer data are considered, and any sorted number with its order in the sorted sequence must be correctly recalled from its compact group.

## Background

*Streaming data sort* [34] proposed a sorting algorithm for large streaming data by representing sorted sub-sequences in forms of compact groups so that the sorting process could be executed under a limited working memory size on a uniprocessor. The compacting process is a major concept of *streaming data sort* to efficiently represent the sorted data set to avoid memory overflow. Four particular sub-sequences of type-1, type-2, type-3, and type-4 are compacted to type-1 compact groups $(u, v)^{(1)}$, type-2 compact groups $(u, v)^{(2)}$, type-3 compact groups $(u, v)^{(3)}$, and type-4 compact groups $(u, v)^{(4)}$, respectively. Type-1 is a sub-sequence of data where all differences between any two consecutive data are one. Type-2 is a sub-sequence of data where the first difference is one and all differences of two consecutive data alternate between one and two. Type-3 is a sub-sequence of data wherein the first difference is two and all differences of two consecutive data alternate between one and two. Finally, type-4 is a sub-sequence

of data wherein all differences of any two consecutive data are two. For examples of the four types compacted to compact groups: (1) a type-1 sub-sequence (14, 15, 16, 17, 18) is compacted to a type-1 compact group $(14, 18)^{(1)}$ in which (15, 16, 17) are removed from memory and called compressed data; (2) a type-2 sub-sequence (14, 15, 17, 18, 20) is compacted to a type-2 compact group $(14, 20)^{(2)}$ and (15, 17, 18) are removed from memory; (3) a type-3 sub-sequence (14, 16, 17, 19, 20) is compacted to a type-3 compact group $(14, 20)^{(3)}$ and (16, 17, 19) are removed from memory; (4) a type-4 sub-sequence (14, 16, 18, 20, 22) is compacted to a type-4 compact group $(14, 22)^{(4)}$ and (16, 18, 20) are removed from memory. A compact group of any type is invertible to the sub-sequence of type $p$ where $p$ is 1, 2, 3 and 4. For example, $(14, 20)^{(3)} = (14, 16, 17, 19, 20)$. A result of compressing all data in memory usage at iteration $t$ is called the compact set $Q^{(t)}$, which contains two member types: compact group and single data. Note that type-$p$ of any sub-sequence $(w_i, w_{i+1}, w_{i+2}, \ldots, w_{i+l})$ can be computed by

$$p = w_{i+2} + w_{i+1} - 2(w_i + 1) \ \text{ where } \ i \in \mathbb{N}. \tag{1}$$

To create $Q^{(t+1)}$, the new incoming data in the $(t + 1)^{th}$ iteration are investigated together with single data and inversion of the compact group in $Q^{(t)}$. The inversion of the compact group can be generated by the recalling process. Inserting incoming data into $Q^{(t)}$ affects the sorting pattern of single data and compact groups in $Q^{(t+1)}$. Fig 1 illustrates an example of 20 simple data points sorted by the *streaming data sort* algorithm, where the memory usage size is 10, which is half of the data size. Note that type-1 and type-4 have higher compression precedence than type-2 and type-3.

## Definitions and notations

Since this paper concerns the improvement of the algorithm in [34], the definitions of introduced sequence types must be recapped here as follows.

**Definition 1 Streaming data sequence** $D = (d_1, d_2, d_3, \ldots, d_n)$, *for* $1 \leq n \leq \infty$, *is a finite or infinite sequence of integers to be sorted.*

The streaming data enter the sorting process in the form of a consecutive chunk of data. There are four types of relations among the data in each chunk, which are defined as follows.

**Definition 2** [34] **Type-1** *sub-sequence* $T_1 = (d_i, \ldots, d_{i+l}) \subseteq D$ *is a sequence such that* $\forall d_i$, $d_{i+1} \in T_1$: $|d_i - d_{i+1}| = 1$.

**Definition 3** [34] **Type-2** *sub-sequence* $T_2 = (d_i, \ldots, d_{i+l}) \subseteq D$ *is a sequence such that* $\forall d_{i+a}$, $d_{i+a+1} \in T_2$, $0 \leq a \leq l - 1$: $|d_{i+a} - d_{i+a+1}| = 1$ *when a is even and* $|d_{i+a} - d_{i+a+1}| = 2$ *when a is odd.*

**Definition 4** [34] **Type-3** *sub-sequence* $T_3 = (d_i, \ldots, d_{i+l}) \subseteq D$ *is a sequence such that* $\forall d_{i+a}$, $d_{i+a+1} \in T_2$, $0 \leq a \leq l - 1$: $|d_{i+a} - d_{i+a+1}| = 2$ *when a is even and* $|d_{i+a} - d_{i+a+1}| = 1$ *when a is odd.*

**Definition 5** [34] **Type-4** *sub-sequence* $T_4 = (d_i, \ldots, d_{i+l}) \subseteq D$ *is a sequence such that* $\forall d_i$, $d_{i+1} \in T_1$: $|d_i - d_{i+1}| = 2$.

**Definition 6 Insert position** $ins(\cdot)$ *of the* $i^{th}$ *datum* $d_i$ *into type-p compact group* $(u, v)^{(p)}$ *is the order of location of* $d_i$ *with respect to all integers in the compact group* $(u, v)^{(p)}$. *This position is computed by the following equations:*

$$ins(d_i) = \begin{cases} 2 \left\lfloor \dfrac{d_i - u}{3} \right\rfloor + (d_i - u) \bmod 3 + 1 & for \ \ p = 2, 3 \\[4mm] \left\lceil \dfrac{d_i - u}{3} \right\rceil + 1 & for \ \ p = 4. \end{cases} \tag{2}$$

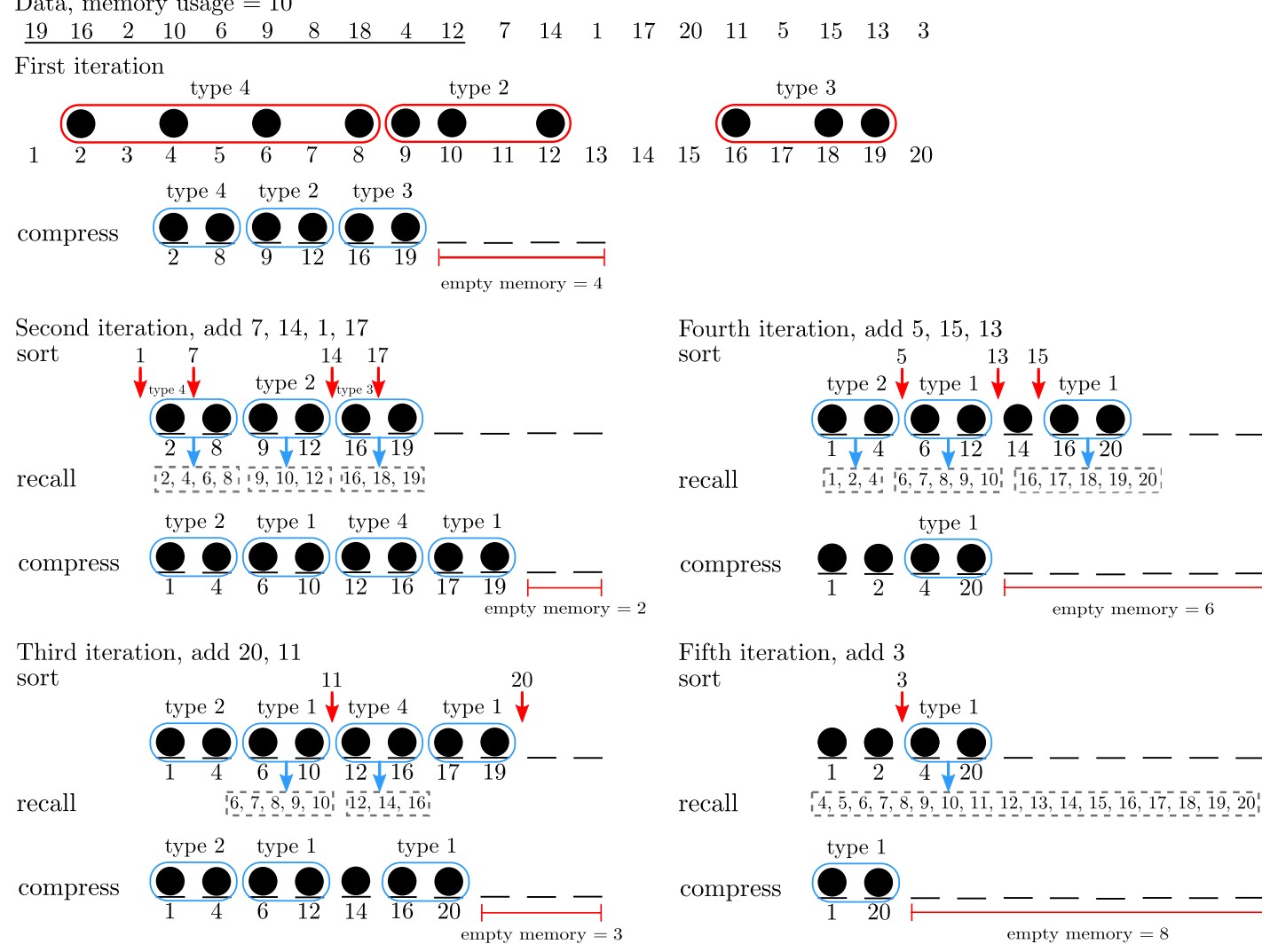

**Fig 1. A *streaming data sort* example.**

**Definition 7 Size of compact group** $(u, v)^{(p)}$ *denoted by* $|(u, v)^{(p)}|$ *is the number of all possible existent integers inside the interval* $(u, v)^{(p)}$ *computed by the following equations*:

$$|(u, v)^{(p)}| = \begin{cases} v - u + 1 & \text{for } p = 1 \\[2mm] 2\left\lfloor \dfrac{v - u}{3} \right\rfloor + (v - u) \bmod 3 & \text{for } p = 2 \\[2mm] \qquad -\dfrac{1}{2}\left\lfloor \dfrac{(v - u) \bmod 3}{2} \right\rfloor + 1 & \\[2mm] 2\left\lfloor \dfrac{v - u}{3} \right\rfloor + \dfrac{(v - u) \bmod 3}{2} + 1 & \text{for } p = 3. \\[2mm] \dfrac{v - u}{2} + 1 & \text{for } p = 4. \end{cases} \tag{3}$$

Notably, if $d_i$ is in $(u, v)^{(p)}$, then $|(u, d_i)^{(p)}|$ is the position of $d_i$ in $(u, v)^{(p)}$.

**Definition 8** *A new incoming datum $d_i$ is a **duplicate** of the already existent $d_i \in (u, v)^{(p)}$ if $d_i$ complies with one of the following conditions.*

1. *If $u \le d_i \le v$ for the type-1 compact group.*

2. *If $u \le d_i \le v$ and $(d_i - u)\mathrm{mod}3 = 0$ or 1 for the type-2 compact group.*

3. *If $u \le d_i \le v$ and $(d_i - u)\mathrm{mod}3 = 0$ or 2 for the type-3 compact group.*

4. *If $u \le d_i \le v$ and $(d_i - u)\mathrm{mod}2 = 0$ for the type-4 compact group.*

Each condition is based on the fact that the distance between two consecutive integers in each compact group must comply with the definitions 2, 3, 4, and 5. Thus, if a compact group already contains $d_i$, then another new incoming $d_i$ must be obviously congruent with the distance defined in either 2, 3, 4, or 5. To denote a datum $d_i$ with no duplicate and a datum with some duplicates, a superscript (*f*) is attached to $d_i$ as $d_i^{(f)}$, for $f \ge 1$, to count the duplicates of $d_i$ in the corresponding compact group. Obviously, $d_i^{(1)}$ implies that $d_i$ has no duplicate.

## Concepts

*Streaming data sort* [34] updates the present set of compact groups when a new incoming datum $d_\alpha$ enters the sorting process by unfolding the relevant compact groups and rearranging the data with $d_\alpha$. After unfolding and rearranging data of the relevant compact group, a new set of compact groups is updated next, possibly by merging or decomposing the present compact groups. Obviously, these steps lengthen the sorting time. Another issue not considered by *Streaming data sort* is data duplication. Unlike *Streaming data sort*, the steps of unfolding and rearranging compact groups are eliminated. New steps are proposed by deploying Definition 6 with Definition 8 to speed up the sorting process as well as to handle data duplication.

Suppose that the working memory size is set to *m* and that there is only one incoming datum $d_\alpha$ at a time. Datum $d_\alpha$ is either inserted into one of the present compact groups or becomes a single number by itself if it cannot be inserted into any compact group. At any time *t*, there are two sets of interest: a set of compact groups called the *compact set* denoted by $Q^{(t)}$ and a set of duplicates $R^{(t)}$. At any time *t*, there exists a new incoming datum $d_\alpha$. The conceptual steps to handle $d_\alpha$ and its duplicates are the followings.

1. Instead of unfolding the concerned compact group to obtain all numbers in the compact group, the *insert position* of $d_\alpha$ is directly computed by applying Definition 6.

2. Suppose $d_\alpha$ is inserted into compact group $(u, v)^{(p)}$. Check whether $(u, v)^{(p)}$ is of types 2, 3, or 4, then recursively split $(u, v)^{(p)}$ at position of $d_\alpha$ into smaller compact groups.

3. Test the duplicate condition by using Definition 8. If the condition is satisfied, then update $R^{(t)}$.

4. Merge any adjacent compact groups into one compact group if they have the same subsequence type.

By splitting the current compact group into several smaller compact groups with reference to the position of $d_\alpha$, the sorting algorithm can achieve faster speed than the speed of *Streaming data sort*. Fig 2 illustrates an example of how the concept of *fast streaming data sort* works. There are 21 numbers to be sorted. The size of working memory is set to *m* = 12. The incoming data whose size is at most 12 flow into the working memory one chunk at a time.

The first incoming chunk includes 10 numbers with two duplicates: 2, 7, 2, 10, 6, 9, 8, 4, 6, 12. These data are sorted and compacted into two compact groups of types 1 and 4, and one

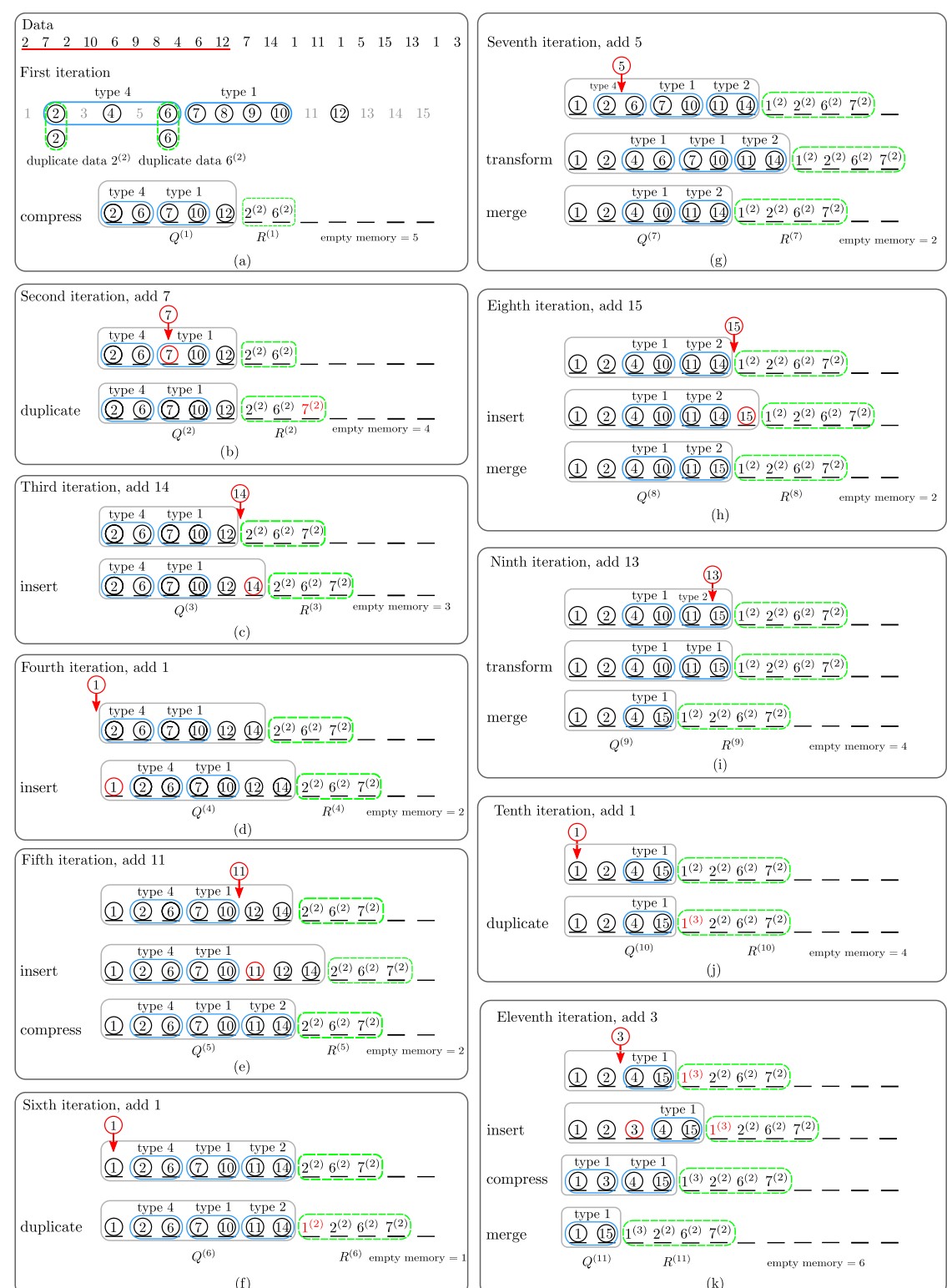

**Fig 2. An example** *fast streaming data sort* **with 11 iterations from (a) to (k).**

single number is assigned to the *compact set* $Q^{(1)} = ((2, 6)^{(4)}, (7, 10)^{(1)}, 12)$ with the duplicate set $R^{(1)} = (2^{(2)}, 6^{(2)})$ as shown in Fig 2(a). The incoming numbers are circled. The less opaque numbers are not currently incoming numbers, but only the numbers in the natural sequence. Notice that only 7 out of 12 entries of memory are used. The second incoming chunk has only one number, 7, which already exists in compact group $(7, 10)^{(1)}$. Datum 7 is a duplicate according to Definition 8. Thus, the present compact set is still the same as before, i.e. $Q^{(2)} = ((2, 6)^{(4)}, (7, 10)^{(1)}, 12)$, but the set of duplicates is updated as $R^{(2)} = (2^{(2)}, 6^{(2)}, \underline{7^{(2)}})$, as illustrated in Fig 2(b).

The third incoming chunk includes only number 14, which can be neither combined with any previous single number to form a new compact group nor assigned to any existing compact groups. This number is inserted behind datum 12. Therefore, $Q^{(3)} = ((2, 6)^{(4)}, (7, 10)^{(1)}, 12, \underline{14})$ and $R^{(3)} = (2^{(2)}, 6^{(2)}, 7^{(2)})$, as shown in Fig 2(c). The fourth incoming number 1 is inserted in front of data 2. The compact set and duplicate set are updated as $Q^{(4)} = (\underline{1}, (2, 6)^{(4)}, (7, 10)^{(1)}, 12, 14)$ and $R^{(4)} = (2^{(2)}, 6^{(2)}, 7^{(2)})$, as shown in Fig 2(d). The fifth incoming number 11 is inserted in front of datum 12 to form a new compact group of type 2, $(11, 14)^{(2)}$. Therefore, $Q^{(5)} = (1, (2, 6)^{(4)}, (7, 10)^{(1)}, (11, 14)^{(2)})$ and $R^{(5)} = (2^{(2)}, 6^{(2)}, 7^{(2)})$, as shown in Fig 2(e). The rest of the incoming numbers shown in Fig 2(f)–2(k) are handled by the same conceptual procedure as previously described. It is remarkable that the memory space used by this sorting process is much less than the available space.

## Proposed *fast streaming data sort* algorithm

The steps to construct each compact group are adopted from the algorithm proposed in [34]. The algorithms to handle duplicate data, merge compact groups, and remove expired data based on the supportive Theorems are the following.

**Main algorithm: Fast streaming data sort algorithm**

```
Input: 1) a chunk of incoming numbers.
Output: (1) Q^(t) and (2) R^(t).
1. Set time t = 1.
2. Obtain the first incoming chunk of numbers
3. Create first compact set Q^(1).
4. Create duplicate set R^(1) using Algorithm 1.
5. t = t + 1.
6. While there exists a new data chunk do.
7.    Insert each d_i into Q^(t-1) using Algorithm 2.
8.    Merge some compact groups if possible.
9.    Update duplicate set R^(t-1) using Algorithm 1.
10.   t = t + 1.
11. EndWhile
```

The details of **Algorithm 1** and **Algorithm 2** are presented in the following sections.

## Handling data duplication

To determine whether $d_i$ entering the sorting algorithm at time $t$ is a duplicate of any previously sorted $d_j$, the following two possible cases are analyzed. The first case is when $d_j$ already exists in the compact set $Q^{(t-1)}$ as a single number. In this case, determining the duplication of $d_i$ is simply performed by comparing $d_i$ with each single number in $Q^{(t-1)}$. The second case occurs when $d_j$ is already covered in a compact group. In this case, the conditions stated in Definition 8 are tested. For example, at time $t - 1$, the following compact set $\{(6, 8)^{(1)}, (9, 12)^{(2)}, (16, 19)^{(3)}, (22, 26)^{(4)}\}$ is formed and this chunk of incoming data 7, 10, 10, 18, 18, 18, 24, 24, 24, 24 enters the sorting algorithm at time $t$. Number 7 is declared as a duplicate of compact group $(6, 8)^{(1)}$ by condition 1 of Definition 8 and number 7 is represented by $7^{(2)}$.

Number 10 is the duplicate of number 10 in compact group $(9, 12)^{(2)}$ because the second condition $(10 − 9)\mod 3 = 1$ is satisfied. Thus, number 10 is represented by $10^{(3)}$. Number 18 is a duplicate of $(16, 19)^{(3)}$ because $(18 − 16)\mod 3 = 2$ by condition 3, and number 24 is a duplicate of $(22, 26)^{(4)}$ because $(24 − 22)\mod 2 = 0$ by condition 4. Both numbers are represented by $18^{(4)}$ and $24^{(4)}$, respectively. The duplicate set $R^{(t)}$ becomes $R^{(t)} = \{7^{(2)}, 10^{(3)}, 18^{(4)}, 24^{(5)}\}$.

Let $d_i^{(*)}$ be a duplicate of $d_i$ kept in duplicate set $R^{(t−1)}$.

**Algorithm 1: Updating duplicate set**

```
Input: (1) dᵢ. (2) Q⁽ᵗ⁻¹⁾, where t ⩾ 2.
Output: (1) R⁽ᵗ⁾.
1. If dᵢ is a duplicate in Q⁽ᵗ⁻¹⁾ then */ by definition 4 /*
2.    If dᵢ⁽*⁾ is in R⁽ᵗ⁻¹⁾ then
3.       Update duplicate count of dᵢ by f = f + 1.
4.    Else
5.       Insert dᵢ in compact set Q⁽ᵗ⁾.
6.    EndIf
7. EndIf
```

## Inserting incoming numbers into compact groups

When a new incoming number is taken into the framework, this number may possibly be one of the numbers in the number sequence of an existing compact group. If the incoming number has type 1, it can be simply inserted at the front or at the rear of the corresponding compact group. Otherwise, some prior process must be performed to form a new compact group or a new set of compact groups by splitting them. Only type-2, type-3, and type-4 require this prior process. When forming a new compact group of types 2, 3, and 4, it does not matter whether the incoming number is a duplicate or not because each compact group just keeps track of the existence of all numbers which are already sorted, and the number of duplicates is instead separately recorded in the duplicate set.

The ways to split compact groups of type-2, type-3, and type-4 *compact groups* into smaller compact groups after inserting a new number are completely different due to the unequal distances between any two consecutive numbers of each type. Furthermore, the size of each compact group measured in terms of the total numbers being compacted in the compact group is also important for splitting the compact group. There are two trivial cases for which compact groups cannot be split. The first case is when the size of the type-2 *compact group* is either 3 or 4. The second case is when the size of the type-3 *compact group* is 3. For other non-trivial cases, the type of compact group can be changed to another type after the insertion as the result of distance change between two consecutive numbers in the compact group. Notice that both type-2 and type-3 *compact groups* become type-1 *compact groups* after insertion. For example, suppose that an incoming number 16 is inserted into type-2 *compact group* $(14, 18)^{(2)}$ of size $|(14, 18)^{(2)}| = 4$. After insertion of 16 into $(14, 18)^{(2)}$, the compact group becomes $(14, 18)^{(1)}$ because the distance between two consecutive numbers is 1. If incoming datum 15 is inserted into type-3 *compact group* $(14, 17)^{(3)}$, then $(14, 17)^{(3)}$ becomes $(14, 17)^{(1)}$. Obviously, the type-2 and type-3 compact groups are always transformed to type-1 *compact groups* after inserting incoming $d_i$

In the general case, the results of inserting an incoming datum into type-2, type-3, and type-4 *compact groups* can be split into smaller compact groups which are classified into two categories: (1) the type-1 *compact group* and (2) the non-type-1 part which may be either a single number or a mixture of *compact group* of type-2, type-3, and type-4. For example, an incoming 16 is inserted into $(11, 17)^{(2)}$. If the size of $|(11, 17)^{(2)}| = 5 > 4$, which is not a trivial case, and $ins(16) = 5$, then the type-1 part is $(14, 17)^{(1)}$ and the non-type-1 part consists of two

single data: 11 and 12. Therefore, $(11, 17)^{(2)}$ transforms to 11, 12, $(14, 17)^{(1)}$. For another example, an incoming 12 is inserted into $(11, 19)^{(3)}$. If $|(11, 17)^{(3)}| = 5 > 4$, not a trivial case, and $ins(12) = 2$, then the type-1 part is $(11, 14)^{(1)}$ and the non-type-1 part is $(16, 19)^{(3)}$. Therefore, $(11, 17)^{(3)}$ transforms to $(11, 14)^{(1)}$, $(16, 19)^{(3)}$.

**Theorem 1** *A non-type-1 part containing single data is generated after inserting an incoming number $d_j$ into a type-2 compact group $(u, v)^{(2)}$, if $ins(d_i)$ is equal to one of the following positions $\{5, |(u, v)^{(2)}| - 3, |(u, v)^{(2)}| - 2\}$.*

**Proof**: There are three cases to be considered, depending on $ins(d_i)$ of the type-2 *compact group* $(u, v)^{(2)}$, as follows.

**Case 1**: $ins(d_i) = 5$. Let $(u, v)^{(2)} = d_i - 5, d_i - 4, d_i - 2, d_i - 1, d_i + 1, d_i + 2, d_i + 4, \ldots, v$ where $u = d_i - 5$. $d_i$ is inserted between $d_i - 1$ and $d_i + 1$. Then, sub-sequence $d_i - 2, d_i - 1, d_i + 1, d_i + 2$ is transformed to type-1 *compact group* $(d_i - 2, d_i + 2)^{(1)}$. Since $u = d_i - 5$ and $u + 1 = d_i - 4$, $u$ and $u - 1$ are single data in the non-type-1 part and sub-sequence $d_i + 4, \ldots, v$ is transformed to type-2 *compact group* $(d_i + 4, v)^{(2)}$. After insertion, $(u, v)^{(2)}$ is transformed to $u, u + 1, (d_i - 2, d_i + 2)^{(1)}, (d_i + 4, v)^{(2)}$.

**Case 2**: $ins(d_i) = |(u, v)^{(2)}| - 3$. Let $(u, v)^{(2)} = u, \ldots, d_i - 4, d_i - 2, d_i - 1, d_i + 1, d_i + 2, d_i + 4, d_i + 5$, where $v = d_i + 5$. $d_i$ is inserted between $d_i - 1$ and $d_i + 1$. Then, sub-sequence $d_i - 2, d_i - 1, d_i, d_i + 1, d_i + 2$ is transformed to type-1 *compact group* $(d_i - 2, d_i + 2)^{(1)}$. Since $v = d_i + 5$ and $v - 1 = d_i + 4$, $v - 1$ and $v$ are single data in the non-type-1 part and sub-sequence $u, \ldots, d_i - 4$ is transformed to type-2 *compact group* $(u, d_i - 4)^{(2)}$. After insertion, $(u, v)^{(2)}$ is transformed to $(u, d_i - 4)^{(2)}, (d_i - 2, d_i + 2)^{(1)}, v - 1, v$.

**Case 3**: $ins(d_i) = |(u, v)^{(2)}| - 2$. Let $(u, v)^{(2)} = u, \ldots, d_i - 4, d_i - 2, d_i - 1, d_i + 1, d_i + 2, d_i + 4$, where $v = d_i + 4$. $d_i$ is inserted between $d_i - 1$ and $d_i + 1$. Then, sub-sequence $d_i - 2, d_i - 1, d_i, d_i + 1, d_i + 2$ is transformed to type-1 *compact group* $(d_i - 2, d_i + 2)^{(1)}$. Since $v = d_i + 4$, $v$ is a single datum in the non-type-1 part and sub-sequence $u, \ldots, d_i - 4$ is transformed to type-2 *compact group* $(u, d_i - 4)^{(2)}$. After insertion, $(u, v)^{(2)}$ is transformed to $(u, d_i - 4)^{(2)}, (d_i - 2, d_i + 2)^{(1)}, v$.

**Theorem 2** *The non-type-1 part is generated after inserting an incoming number $d_j$ into a type-3 compact group $(u, v)^{(3)}$, if $ins(d_i)$ is equal to one of the following positions $\{4, |(u, v)^{(3)}| - 3, |(u, v)^{(3)}| - 2\}$.*

**Proof**: This proof is similar to **Theorem 1**.

**Case 1**: When $ins(d_i) = 4$, then $(u, v)^{(3)} = d_i - 4, d_i - 2, d_i - 1, d_i + 1, d_i + 2, d_i + 4, \ldots, v$ where $u = d_i - 4$. Sub-sequences $d_i - 2, d_i - 1, d_i, d_i + 1, d_i + 2$ and $d_i + 4, \ldots, v$ are transformed to $(d_i - 2, d_i + 2)^{(1)}$ and $(d_i + 4, v)^{(2)}$, respectively. $u$ is a single datum in the non-type-1 part. After insertion, $(u, v)^{(3)}$ is transformed to $u, (d_i - 2, d_i + 2)^{(1)}, (d_i + 4, v)^{(2)}$.

**Case 2**: $ins(d_i) = |(u, v)^{(3)}| - 3$. Then, $(u, v)^{(3)} = u, \ldots, d_i - 4, d_i - 2, d_i - 1, d_i + 1, d_i + 2, d_i + 4, d_i + 5$, where $v = d_i + 5$. Sub-sequences $d_i - 2, d_i - 1, d_i, d_i + 1, d_i + 2$ and $u, \ldots, d_i - 4$ are transformed to $(d_i - 2, d_i + 2)^{(1)}$ and $(u, d_i - 4)^{(3)}$, respectively. $v - 1$ and $v$ are single data in the non-type-1 part. After insertion, $(u, v)^{(3)}$ is transformed to $(u, d_i - 4)^{(3)}, (d_i - 2, d_i + 2)^{(1)}, v - 1, v$.

**Case 3**: $ins(d_i) = |(u, v)^{(3)}| - 2$. Then, $(u, v)^{(3)} = u, \ldots, d_i - 4, d_i - 2, d_i - 1, d_i + 1, d_i + 2, d_i + 4$, where $v = d_i + 4$. Sub-sequences $d_i - 2, d_i - 1, d_i, d_i + 1, d_i + 2$ and $u, \ldots, d_i - 4$ are transformed to $(d_i - 2, d_i + 2)^{(1)}$ and $(u, d_i - 4)^{(3)}$, respectively. $v$ is a single datum in the non-type-1 part. After insertion, $(u, v)^{(3)}$ is transformed to $(u, d_i - 4)^{(3)}, (d_i - 2, d_i + 2)^{(1)}, v$.

**Theorem 3** *The non-type-1 part is generated after inserting an incoming datum $d_j$ into a type-4 compact group $(u, v)^{(4)}$, if $ins(d_i)$ is equal to one of the following positions $\{3, |(u, v)^{(4)}| - 2, |(u, v)^{(4)}| - 1\}$.*

**Proof**: This proof is similar to **Theorem 1**.

**Table 2. Summary of insertion position and the sequence of insertion in forms of split compact groups.**

| Type-$p$ | Insert position $ins(d_i)$ | new resultant split compact groups $(u, v)^{(p)}$ |
|---|---|---|
| 2 (Trivial[1]) | 3 | $(u, v)^{(1)}$ |
| 2 | 5 | $u, u + 1, (d_i - 2, d_i + 2)^{(1)}, (d_i + 4, v)^{(2)}$ |
| 2 | $|(u, v)^{(2)}| - 3$ | $(u, d_i - 4)^{(2)}, (d_i - 2, d_i + 2)^{(1)}, v - 1, v$ |
| 2 | $|(u, v)^{(2)}| - 2$ | $(u, d_i - 4)^{(2)}, (d_i - 2, d_i + 2)^{(1)}, v$ |
| 2 | others | $(u, d_i - 4)^{(2)}, (d_i - 2, d_i + 2)^{(1)}, (d_i + 4, v)^{(2)}$ |
| 3 (Trivial[2]) | 2 | $(u, v)^{(1)}$ |
| 3 | 4 | $u, (d_i - 2, d_i + 2)^{(1)}, (d_i + 4, v)^{(2)}$ |
| 3 | $|(u, v)^{(3)}| - 3$ | $(u, d_i - 4)^{(3)}, (d_i - 2, d_i + 2)^{(1)}, v - 1, v$ |
| 3 | $|(u, v)^{(3)}| - 2$ | $(u, d_i - 4)^{(3)}, (d_i - 2, d_i + 2)^{(1)}, v$ |
| 3 | others | $(u, d_i - 4)^{(3)}, (d_i - 2, d_i + 2)^{(1)}, (d_i + 4, v)^{(2)}$ |
| 4 | 3 | $u, (d_i - 1, d_i + 1)^{(1)}, (d_i + 3, v)^{(4)}$ |
| 4 | $|(u, v)^{(4)}| - 2$ | $(u, d_i - 3)^{(4)}, (d_i - 1, d_i + 1)^{(1)}, v - 2, v$ |
| 4 | $|(u, v)^{(4)}| - 1$ | $(u, d_i - 3)^{(4)}, (d_i - 1, d_i + 1)^{(1)}, v$ |
| 4 | others | $(u, d_i - 3)^{(4)}, (d_i - 1, d_i + 1)^{(1)}, (d_i + 3, v)^{(4)}$ |

Trivial[1] refers to the case in which the size of the type-2 compact group is either 3 or 4. Trivial[2] refers to the case in which the size of the type-3 compact group is 3.

**Case 1**: $ins(d_i) = 3$, and then $(u, v)^{(4)} = d_i - 3, d_i - 1, d_i + 1, d_i + 3, \ldots, v$ where $u = d_i - 3$. Sub-sequences $d_i - 1, d_i, d_i + 1$ and $d_i + 3, \ldots, v$ are transformed to $(d_i - 1, d_i + 1)^{(1)}$ and $(d_i + 3, v)^{(4)}$, respectively. $u$ is a single datum in the non-type-1 part. After insertion, $(u, v)^{(4)}$ is transformed to $u, (d_i - 1, d_i + 1)^{(1)}, (d_i + 3, v)^{(4)}$.

**Case 2**: $ins(d_i) = |(u, v)^{(4)}| - 2$, then $(u, v)^{(4)} = u, \ldots, d_i - 3, d_i - 1, d_i + 1, d_i + 3, d_i + 5$, where $v = d_i + 5$. Sub-sequences $d_i - 1, d_i, d_i + 1$ and $u, \ldots, d_i - 3$ are transformed to $(d_i - 1, d_i + 1)^{(1)}$ and $(u, d_i - 3)^{(4)}$, respectively. $v - 2$ and $v$ are single data in the non-type-1 part. After insertion, $(u, v)^{(4)}$ is transformed to $(u, d_i - 3)^{(4)}, (d_i - 1, d_i + 1)^{(1)}, v - 2, v$.

**Case 3**: $ins(d_i) = |(u, v)^{(4)}| - 1$, then $(u, v)^{(4)} = u, \ldots, d_i - 3, d_i - 1, d_i + 1, d_i + 3$ where $v = d_i + 3$. Sub-sequence $d_i - 1, d_i, d_i + 1$ and $u, \ldots, d_i - 3$ are transformed to $(d_i - 1, d_i + 1)^{(1)}$ and $(u, d_i - 3)^{(4)}$, respectively. $v$ is a single datum in the non-type-1 part. After insertion, $(u, v)^{(4)}$ is transformed to $(u, d_i - 3)^{(4)}, (d_i - 1, d_i + 1)^{(1)}, v$.

If $ins(d_i)$ does not conform with *Theorems 1, 2*, and *3*, then a set of compact groups is generated without the existence of any single number as follows. Type-2 *compact group* $(u, v)^{(2)}$ is transformed to $(u, d_i - 4)^{(2)}, (d_i - 2, d_i + 2)^{(1)}, (d_i + 4, v)^{(2)}$. Type-3 *compact group* $(u, v)^{(3)}$ is transformed to $(u, d_i - 4)^{(3)}, (d_i - 2, d_i + 2)^{(1)}, (d_i + 4, v)^{(2)}$. Type-4 *compact group* $(u, v)^{(4)}$ is transformed to $(u, d_i - 3)^{(4)}, (d_i - 1, d_i + 1)^{(1)}, (d_i + 3, v)^{(4)}$. Table 2 summarizes the splitting of type-2, type-3 and type-4 *compact groups* as a result of inserting $d_i$. The steps to generate a new compact set after insertion are provided in the following **Algorithm 2**. Let $(u, v)^{(p)}$ denote the corresponding compact group of type-$p$ for inserting the incoming $d_i$.

**Algorithm 2: Generating a new compact set after insertion**

```
Input: (1) dᵢ and (2) compact set Q^(t-1), where t ⩾ 2.
Output: (1) a new set of compact groups Q^(t).
1. Find the corresponding compact group (u, v)^(p) where p ∈ {2, 3, 4}
   for inserting dᵢ.
2. Case:
3.   1: dᵢ is in (u, v)^(2)
```

```
4.      If |(u, v)⁽²⁾| = 3 then /*trivial case*/
5.        Split (u, v)⁽²⁾ into (u, v)⁽¹⁾.
6.      ElseIf ins(dᵢ) = 5
7.        Split (u, v)⁽²⁾ into u, u + 1, (dᵢ − 2, dᵢ + 2)⁽¹⁾, (dᵢ + 4,
          v)⁽²⁾.
8.      ElseIf ins(dᵢ) = |(u, v)⁽²⁾| − 3
9.        Split (u, v)⁽²⁾ into (u, dᵢ − 4)⁽²⁾, (dᵢ − 2, dᵢ + 2)⁽¹⁾, v − 1,
v.
10.     ElseIf ins(dᵢ) = |(u, v)⁽²⁾| − 2
11.       Split (u, v)⁽²⁾ into (u, dᵢ − 4)⁽²⁾, (dᵢ − 2, dᵢ + 2)⁽¹⁾, v.
12.     Else
13.       Split (u, v)⁽²⁾ into (u, dᵢ − 4)⁽²⁾, (dᵢ − 2, dᵢ + 2)⁽¹⁾, (dᵢ
          + 4, v)⁽²⁾.
14.     End
15.   2: dᵢ is in (u, v)⁽³⁾.
16.     If |(u, v)⁽³⁾| = 3 then /*trivial case*/
17.       Split (u, v)⁽³⁾ into (u, v)⁽¹⁾.
18.     ElseIf ins(dᵢ) = 4
19.       Split (u, v)⁽³⁾ into u, (dᵢ − 2, dᵢ + 2)⁽¹⁾, (dᵢ + 4, v)⁽²⁾.
20.     ElseIf ins(dᵢ) = |(u, v)⁽³⁾| − 3
21.       Split (u, v)⁽³⁾ into (u, dᵢ − 4)⁽³⁾, (dᵢ − 2, dᵢ + 2)⁽¹⁾, v − 1,
v.
22.     ElseIf ins(dᵢ) = |(u, v)⁽³⁾| − 2
23.       Split (u, v)⁽³⁾ into (u, dᵢ − 4)⁽³⁾, (dᵢ − 2, dᵢ + 2)⁽¹⁾, v.
24.     Else
25.       Split (u, v)⁽³⁾ into (u, dᵢ − 4)⁽³⁾, (dᵢ − 2, dᵢ + 2)⁽¹⁾, (dᵢ
          + 4, v)⁽²⁾.
26.     EndIf
27.   3: dᵢ is in (u, v)⁽⁴⁾.
28.     If |(u, v)⁽⁴⁾| = 3 then
29.       Split (u, v)⁽⁴⁾ into u, (dᵢ − 1, dᵢ + 1)⁽¹⁾, (dᵢ + 3, v)⁽⁴⁾.
30.     ElseIf ins(dᵢ) = |(u, v)⁽⁴⁾| − 2
31.       Split (u, v)⁽⁴⁾ into (u, dᵢ − 3)⁽⁴⁾, (dᵢ − 1, dᵢ + 1)⁽¹⁾, v − 2,
v.
32.     ElseIf ins(dᵢ) = |(u, v)⁽⁴⁾| − 1
33.       Split (u, v)⁽⁴⁾ into (u, dᵢ − 3)⁽⁴⁾, (dᵢ − 1, dᵢ + 1)⁽¹⁾, v.
34.     Else
35.       Split (u, v)⁽⁴⁾ into (u, dᵢ − 3)⁽⁴⁾, (dᵢ − 1, dᵢ + 1)⁽¹⁾, (dᵢ
          + 3, v)⁽⁴⁾.
36.     EndIf
37. EndCase
```

## Merging compact groups

After compact group splitting, it is possible that some compact group $(u, v)^{(p)}$ can be merged with a single number $s$ to reduce the consumed memory space. The merging conditions of all types of compact groups are the following.

1. If $u - s = 1$ and $p = 1$, then $s$ and $(u, v)^{(p)}$ are merged into $(s, v)^{(1)}$.

2. If $u - s = 2$ and $p = 2$, then $s$ and $(u, v)^{(p)}$ are merged into $(s, v)^{(3)}$.

3. If $u - s = 1$ and $p = 3$, then $s$ and $(u, v)^{(p)}$ are merged into $(s, v)^{(2)}$.

4. If $u - s = 2$ and $p = 4$, then $s$ and $(u, v)^{(p)}$ are merged into $(s, v)^{(4)}$.

5. If $s - v = 1$ and $p = 1$, then $(u, v)^{(p)}$ and $s$ are merged into $(u, s)^{(1)}$.

6. If $s - v = (v - u)\bmod 3 + 1 \leqslant 2$, and $p = 2$, then $(u, v)^{(p)}$ and $s$ are merged into $(u, s)^{(2)}$.

7. If $s - v = (v - u + 2)\bmod 3 \leqslant 2$, and $p = 3$, then $(u, v)^{(p)}$ and $s$ are merged into $(u, s)^{(3)}$.

8. If $s - v = 2$ and $p = 4$, then $(u, v)^{(p)}$ and $s$ are merged into $(u, s)^{(4)}$.

Another possibility is that two *compact groups* can be merged into one *compact group*. The conditions for merging any two *compact groups* $(u_1, v_1)^{(p_1)}$ and $(u_2, v_2)^{(p_2)}$, where $v_1 < u_2$, are defined as follows.

1. If $u_2 - v_1 = 1$ and $p_1 = p_2 = 1$, then $(u_1, v_1)^{(p_1)}$ and $(u_2, v_2)^{(p_2)}$ are merged into $(u_1, v_2)^{(1)}$.

2. If $u_2 - v_1 = 2 = (v_1 - u_1 + 1)\bmod 3$ and $p_1 = p_2 = 2$, then $(u_1, v_1)^{(p_1)}$ and $(u_2, v_2)^{(p_2)}$ are merged into $(u_1, v_2)^{(2)}$.

3. If $u_2 - v_1 = 1 = (v_1 - u_1 + 1)\bmod 3$, $p_1 = 2$, and $p_2 = 3$, then $(u_1, v_1)^{(p_1)}$ and $(u_2, v_2)^{(p_2)}$ are merged into $(u_1, v_2)^{(2)}$.

4. If $u_2 - v_1 = 2 = (v_1 - u_1 + 2)\bmod 3$, $p_1 = 3$, and $p_2 = 2$, then $(u_1, v_1)^{(p_1)}$ and $(u_2, v_2)^{(p_2)}$ are merged into $(u_1, v_2)^{(3)}$.

5. If $u_2 - v_1 = 1 = (v_1 - u_1 + 2)\bmod 3$ and $p_1 = p_2 = 3$, then $(u_1, v_1)^{(p_1)}$ and $(u_2, v_2)^{(p_2)}$ are merged into $(u_1, v_2)^{(3)}$.

6. If $u_2 - v_1 = 2$ and $p_1 = p_2 = 4$, then $(u_1, v_1)^{(p_1)}$ and $(u_2, v_2)^{(p_2)}$ are merged into $(u_1, v_2)^{(4)}$.

For example, the result of merging $(4, 8)^{(2)}$ and $(10, 12)^{(2)}$ is $(4, 12)^{(2)}$ because $10 - 8 = 2 = (8 - 4 + 1)\bmod 3$. Note that merging two *compact groups* depends on the order of *compact groups* in a *compact set*.

## Removing expired single numbers from the compact set

Various applications in streaming data mining and data life cycle management must occasionally remove some expired data from the storage or database. Actually, removing any data from a sorted sequence when the whole sequence is kept in storage is rather simple. However, in cases of sequences sorted with limited storage size, removing any data from the sorted sequence is not as simple because the whole sorted sequence is transformed into a set of compact groups and single numbers. *Streaming data sort* [34] does not support data removal from a compact set during the sorting process.

The process of data removal from a compact set is proposed in this *fast streaming data sort*. A removed datum is denoted by $\tilde{d}_i$, while a set of deleted data is denoted by $\tilde{D}$. There are two removal cases. The first case is that $\tilde{d}_i$ is a single number in a compact set. In this case, $\tilde{d}_i$ can be directly removed from the compact set without affecting the components of other compact groups. For the second case, $\tilde{d}_i$ is a member of a *compact group* $(u, v)^{(p)}$. Removing $\tilde{d}_i$ from $(u, v)^{(p)}$ splits $(u, v)^{(p)}$ into two parts: the left part $L$ and right part $R$ with respect to $\tilde{d}_i$. Depending upon the type of compact group, the set of numbers in parts $L$ and $R$ can be expressed by the following Theorems.

**Theorem 4** *Let L be a set of a left part for removing a datum $\tilde{d}_i$ from a compact group* $(u, v)^{(p)}$. *If* $|(u, \tilde{d}_i)^{(p)}| \in \mathbb{Z}$ *and* $|(u, \tilde{d}_i)^{(p)}| \leq 3$, *then*

$$L = \{l \in \{u, u + 1, u + 2\} \mid |(u, l)^{(p)}| \in \mathbb{Z}\}. \tag{4}$$

**Theorem 5** *Let R be a set of a right part for removing a datum $\tilde{d}_i$ from a compact group* $(u, v)^{(p)}$. *If* $|(u, \tilde{d}_i)^{(p)}| \in \mathbb{Z}$ *and* $|(u, v)^{(p)}| - |(u, \tilde{d}_i)^{(p)}| \leq 2$, *then*

$$R = \{r \in \{v - 2, v - 1, v\} \mid |(u, r)^{(p)}| \in \mathbb{Z}\}. \tag{5}$$

**Theorem 6** *Let L be the left part for removing a datum $\tilde{d}_i$ from a compact group $(u, v)^{(p)}$. If $|(u, \tilde{d}_i)^{(p)}| \in \mathbb{Z}$ and $|(u, \tilde{d}_i)^{(p)}| > 3$, then L has one of the following compact group forms*:

$$L = \begin{cases} (u, \tilde{d}_i - 1)^{(1)} & \text{for } p = 1 \\ (u, \tilde{d}_i - a)^{(2)} & \text{for } p = 2 \\ (u, \tilde{d}_i - b)^{(3)} & \text{for } p = 3 \\ (u, \tilde{d}_i - 2)^{(4)} & \text{for } p = 4 \end{cases} \tag{6}$$

*where $a = 1 + |(u, \tilde{d}_i)^{(p)}| \bmod 2$ and $b = 1 + (|(u, \tilde{d}_i)^{(p)}| + 1) \bmod 2$.*

**Theorem 7** *Let R be a right part for removing a datum $\tilde{d}_i$ from a compact group $(u, v)^{(p)}$. If $|(u, \tilde{d}_i)^{(p)}| \in \mathbb{Z}$ and $|(u, v)^{(p)}| - |(u, \tilde{d}_i)^{(p)}| \le 2$, then R has one of the following compact group forms*:

$$R = \begin{cases} (\tilde{d}_i + 1, v)^{(1)} & \text{for } p = 1 \\ (\tilde{d}_i + a, v)^{(4-a)} & \text{for } p = 2 \\ (\tilde{d}_i + b, v)^{(4-b)} & \text{for } p = 3 \\ (\tilde{d}_i + 2, v)^{(4)} & \text{for } p = 4 \end{cases} \tag{7}$$

*where $a = 2 - |(u, \tilde{d}_i)^{(p)}| \bmod 2$ and $b = 2 - (|(u, \tilde{d}_i)^{(p)}| + 1) \bmod 2$.*

For example, number 25 is removed from $(23, 31)^{(1)}$. By Theorem 4, $|(23, 25)^{(1)}| \le 3$ and the numbers in *L* are {23, 24}. For *R*, $|(23, 31)^{(1)}| - |(23, 25)^{(1)}| > 2$ and $R = (26, 31)^{(1)}$ by Theorem 7. Therefore, removing 25 from $(23, 31)^{(1)}$ splits the compact group into $L = \{23, 24\}$ and $R = (26, 31)^{(1)}$. Another example is removing number 26 from $(23, 31)^{(3)}$. Since $|(23, 26)^{(3)}| \le 3$ and $|(23, 31)^{(3)}| - |(23, 25)^{(3)}| > 2$, the numbers in *L* are {23, 25} by Theorem 4 and $R = (28, 31)^{(2)}$ by Theorem 7. Therefore, the compact group $(23, 31)^{(3)}$ is split by number 26 into $L = \{23, 25\}$ and $R = (28, 31)^{(2)}$. Notice that, with the terms of $4 - a$ and $4 - b$ in Theorem 7, the type of *compact group* after removing $\tilde{d}_i$ will be changed from type 2 to type 3 and vice versa.

## Experimental results and discussion

Three important issues concerning the performance of the proposed algorithm were evaluated. The first issue regards the execution time of the proposed sorting algorithm versus the working memory size in terms of percentage of total experimented data size. The second issue focuses on the fluctuation of working memory space with respect to the given number of numbers to be sorted. The fluctuation indicates the bound of the actual size of working memory required to achieve the sorting process. The third issue concerns the execution time when there are duplicates. The performance of the *fast streaming data sorting algorithm* was compared with *streaming data sort*, classic sorting algorithms, and external merge sort. The proposed algorithm was implemented by MATLAB R2021a and run on 3.2 GHz Intel Core i5 6500 and 8 GB of 2133 MHz RAM with the 64 bit Windows 10 platform. The details of each issue are the following.

### Sorting time versus data size and working memory size

One million random numbers were sorted by the proposed algorithm and the sorting time was evaluated by varying the size of working memory in terms of percentage of total experimented data size. Different sorting time versus different working memory size is illustrated in Fig 3. There are two experiments conducted in this section. The first experiment was

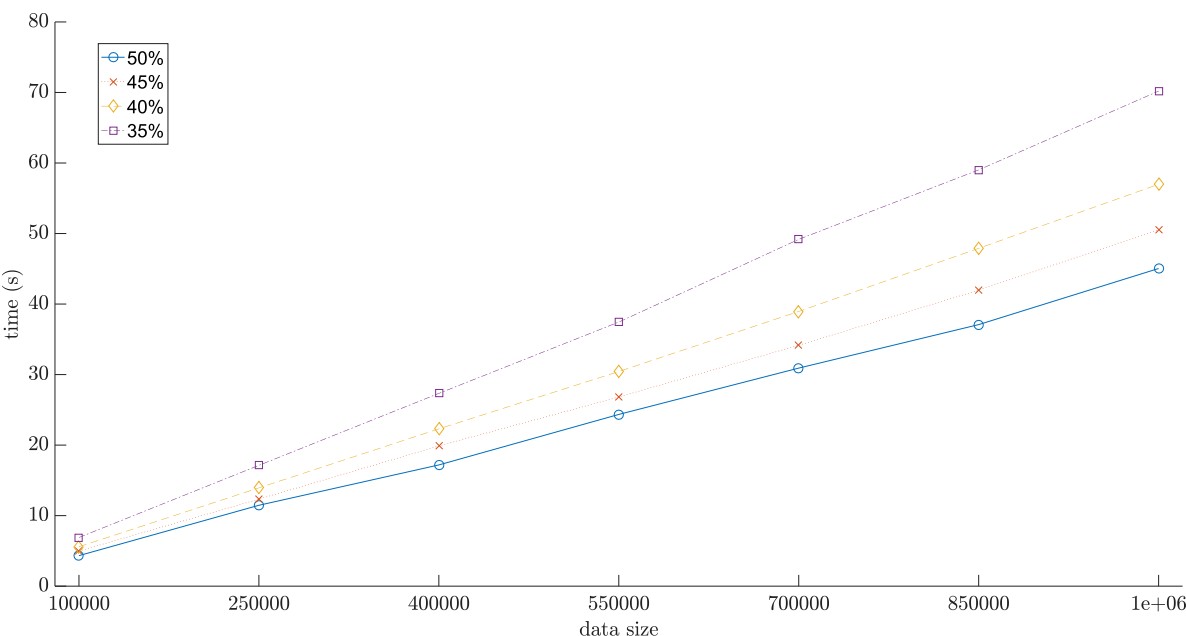

**Fig 3. The execution time of *fast streaming data sort* with respect to the setting of working memory size in terms of percentage of total data size (1,000,000 numbers).**

conducted to observe whether or not the sorting time increases almost linearly according to the accumulated amount of incoming numbers when the size of working memory is fixed. Each line in Fig 3 denotes the sorting time versus the size of the incoming chunk based on a fixed working memory size. The second experiment was conducted to observe the effect of working memory size on the sorting time. It is remarkable that when the size of working memory increases, the sorting is reduced. For example, the sorting time when the size of working memory is 50% of the total data size is less than the sorting time when the size of working memory is 30% of the total data size. This is because more incoming numbers can enter the sorting process, thus causing the sorting time to decrease.

The sorting speed of the *fast streaming data sort* was compared with the sorting speed of the *streaming data sort*, which is capable of sorting streaming data. Table 3 shows the comparison of execution time between *fast streaming data sort* and *streaming data sort*. Each percentage number denotes the size of working memory in terms of the percentage of the total amount of data to be sorted. Each bold number in each gray cell is the sorting time of *fast streaming data sort*, which is obviously 28.1767 times faster than the sorting time of *streaming data sort* on average. The time complexity of *fast streaming data sort* is still $O(n)$.

Furthermore, the sorting speed of *fast streaming data sort* was also compared with the speeds of many classic sort algorithms. Although this comparison is inappropriate because of the distinguished constraints imposed by *fast streaming data sort* and classic sorting algorithms, this comparison can be fairly justified when the size of working memory approaches the size of the data set. For classic sorting algorithms, the entire data set must be stored inside the working memory, which may refer to only internal storage or the combination of internal and external storage, during the sorting process. Table 4 shows the sorting time of the proposed algorithm and four other classic sorting algorithms, i.e. bubble sort, quick sort, insertion sort, and merge sort. The working memory size of the proposed algorithm is restricted to 35%, 50%, 75%, and 98.5% of data size. Notice that when the size of working memory is increased, it

**Table 3. Comparison of sorting time between *fast streaming data sort* and *streaming data sort* when varying different sizes of working memory in terms of the percentages of total amount of data.**

| Data size | Time (sec) of fast streaming data sort | | | | Time (sec) of streaming data sort | | | |
|---|---|---|---|---|---|---|---|---|
| | 50% | 45% | 40% | 35% | 50% | 45% | 40% | 35% |
| 100,000 | 4.3338 | 4.9884 | 5.5894 | 6.9066 | 124.8196 | 137.7921 | 160.3303 | 190.7497 |
| 250,000 | 11.4705 | 12.3407 | 13.9405 | 17.1341 | 324.145 | 356.7402 | 422.1132 | 480.4968 |
| 400,000 | 17.1907 | 19.8813 | 22.3061 | 27.3360 | 498.5432 | 547.7929 | 645.2144 | 747.6058 |
| 550,000 | 24.3549 | 26.8283 | 30.4382 | 37.4500 | 680.8660 | 753.8266 | 901.9769 | 1027.8510 |
| 700,000 | 30.9219 | 34.1581 | 38.9551 | 49.1797 | 861.3885 | 958.9641 | 1139.8098 | 1312.4261 |
| 850,000 | 37.0885 | 41.9728 | 47.9061 | 59.0234 | 1051.8819 | 1169.8349 | 1390.2639 | 1612.7994 |
| 1,000,000 | 45.0412 | 50.5456 | 56.9898 | 70.2029 | 1238.8804 | 1379.0245 | 1643.9551 | 1873.0513 |

**Table 4. Comparison of sorting time between *fast streaming data sort* and four classic sorting algorithms.**

| Data size | The execution time (sec) | | | | | | | |
|---|---|---|---|---|---|---|---|---|
| | Fast streaming data sort | | | | Bubble sort | Quick sort | Insertion sort | Merge sort |
| | 35% | 50% | 75% | 98.5% | | | | |
| $10^3$ | 0.3738 | 0.3193 | 0.2345 | 0.1524 | 0.0144 | 0.0222 | 0.0085 | 0.0123 |
| $10^4$ | 0.9601 | 0.6843 | 0.3690 | 0.1756 | 1.3513 | 0.1434 | 0.4739 | 0.0272 |
| $10^5$ | 6.2055 | 3.7647 | 1.9997 | 0.2913 | 126.3189 | 1.4645 | 50.1042 | 0.1494 |
| $10^6$ | 58.1738 | 34.4192 | 17.5550 | 1.4446 | N/A | 19.4219 | N/A | 1.4890 |
| $10^7$ | 593.6722 | 330.6568 | 181.5882 | 13.1480 | N/A | 218.5080 | N/A | 16.1969 |

implies that a larger incoming data chunk size can be stored in the working memory, thus speeding the sorting as illustrated in Fig 3. Each of the underlined numbers in each gray cell of Table 4 represents the execution times of four classic sorting algorithms that are slower than *fast streaming data sort* for some restricted working memory. Regardless of any memory size, the speed of *fast streaming data sort* exceeds those of bubble sort and insertion sort when data size is $1 \times 10^5$ or greater. N/A denotes excessively long processing time. The sorting time of *fast streaming data sort*, when the data size is at least $1 \times 10^6$ and the size of working memory is set to at least 75% of data size, is less than the sorting time of quick sort. However, when the data size is at least $1 \times 10^6$ and the size of working memory is set to 98.5% of the data set, its speed is faster than the speed of merge sort.

Four classic sorting algorithms do not require any extra storage. This means that they are not suitable for coping with large data sets. External sorting resolves this problem by deploying a very large external storage in addition to the internal working memory to store the overflowed data. Thus, it is interesting to compare the sorting speed of *fast streaming data sort* with the speed of external sorting restricted by different sizes of working memory defined in terms of percentage of the entire data size. Table 5 shows the sorting times of *fast streaming data sort* and external merge sort when varying the size of working memory to 35%, 50%, 75%, and

**Table 5. Comparison of sorting times of *fast streaming data sort* and external merge sort.**

| Data size | Time (sec) of fast streaming data sort | | | | Time (sec) of external merge sort | | | |
|---|---|---|---|---|---|---|---|---|
| | 50% | 45% | 40% | 35% | 50% | 45% | 40% | 35% |
| $10^3$ | 0.3738 | 0.3193 | 0.2345 | 0.1524 | 2.6021 | 2.2978 | 2.2204 | 2.1301 |
| $10^4$ | 0.9601 | 0.6843 | 0.3690 | 0.1756 | 38.0198 | 37.4343 | 35.6365 | 35.525 |
| $10^5$ | 6.2055 | 3.7647 | 1.9997 | 0.2913 | 298.8189 | 392.2904 | 417.2926 | 422.9222 |

98.5% of the data size. It is obvious that *fast streaming data sort* is faster than external merge sort in all circumstances. In particular, when the memory size is set to 98.5% of data size and the data set includes $1 \times 10^5$ numbers, the sorting time of *fast streaming data sort* is 1,452 times less than the sorting time of external merge sort.

## Memory usage

The fluctuation of memory usage can be separated into three stages for analysis. In the beginning stage, the size of memory usage depends on the random order of incoming data. The trend of fluctuation is positive. In the second stage, some single data may appear during the sorting process. When new incoming data enter, they may be inserted into some previous *compact groups* or combined with the other single data to generate new *compact groups*. Some of these *compact groups* can be merged if they are in consecutive order. After merging some compact groups, the available memory space increases. The trend of fluctuation thus possibly changes in this stage. In the last stage, almost all early incoming data form partially complete sequences. The new incoming data fulfill the existing compact groups or concatenate with some single data to form new compact groups. Some of these compact groups can be further merged into new compact groups. Thus, the actual size of memory usage is rapidly reduced. Fig 4 illustrates the memory usage for *compact groups* and single data sets at each time step of the proposed algorithm with different sizes of working memory defined in terms of percentage of data size. When the size of working memory increases, the chance to form only one compact group is higher than having small working memory size because of the higher probability of obtaining consecutive numbers. The shape of each fluctuation curve is similar to a capsized hook. This reason for this shape phenomenon was discussed in [34]. There are four curves as a result of setting different sizes of working memory (35%, 40%, 45%, and 50%). It is remarkable

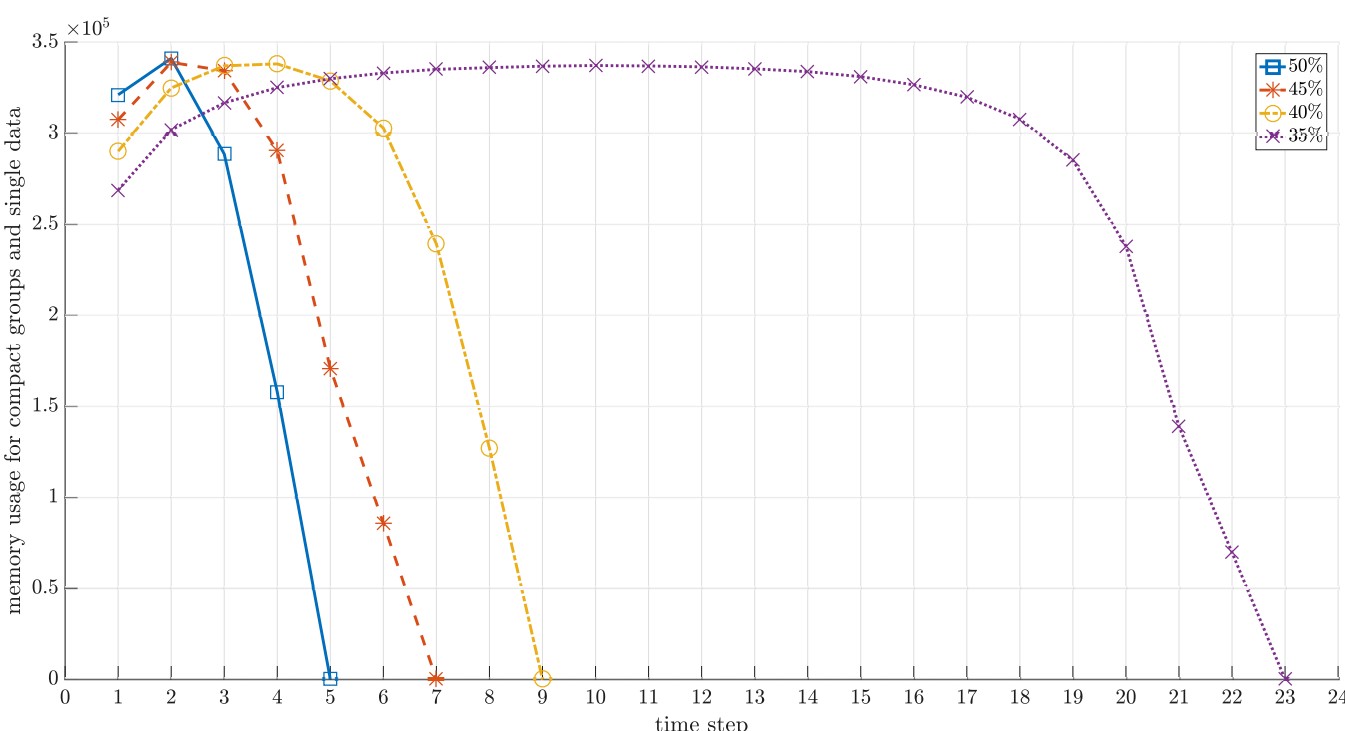

**Fig 4. Fluctuation in number of *compact groups* and single data for the proposed algorithm at several memory sizes for one million data points.**

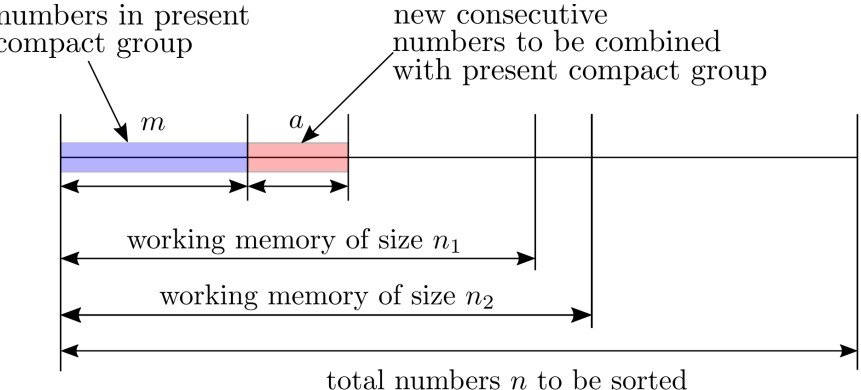

**Fig 5. Scenarios of Theorem 8.** The working memory sizes are $n_1$ and $n_2$.

that a larger size of working memory requires fewer time steps to accomplish the sorting result. The rationale for this behavior can be analyzed in the following Theorem.

**Theorem 8** *When the size of working memory increases, the time steps of the fluctuation curve decrease.*

**Proof**: There are two scenarios in Fig 5 of different sizes of working memory, i.e. $n_1 < n_2$, to be discussed. The same set of incoming $n$ numbers is sorted in both scenarios. The first scenario includes one currently existing compact group containing $m$ numbers. After compacting these $m$ numbers into a compact group, only 3 entries of memory space are required to represent the compact group. Since $3 \ll n_1$, the rest of working memory space is $n_1 - 3 \approx n_1$. Next, a set of consecutive $a \le n_1$ numbers enters the sorting process. These $a$ numbers can be combined with the current compact group to form a new compact group. The rest of the working memory can be filled by selecting $n_1 - a$ from $n - m - a$ numbers. The number of ways to select $n_1 - a$ is equal to $\binom{n - m - a}{n_1 - a}$. All $n_1$ numbers can be permuted in $n_1!$ ways. Thus, the probability of having $a$ numbers in the working memory in the first scenario is as follows.

$$prob_1 = \frac{n_1!\binom{n - m - a}{n_1 - a}}{(n - m)!} \tag{8}$$

$$= \frac{n_1!(n - m - a)!}{(n_1 - a)!(n - m - n_1)!(n - m)!} \tag{9}$$

For the second scenario, the probability of having $a$ numbers in the working memory in the second scenario is as follows.

$$prob_2 = \frac{n_2!\binom{n - m - a}{n_2 - a}}{(n - m)!} \tag{10}$$

$$= \frac{n_2!(n - m - a)!}{(n_2 - a)!(n - m - n_2)!(n - m)!} \tag{11}$$

Since $n_1 < n_2, prob_1 < prob_2$.

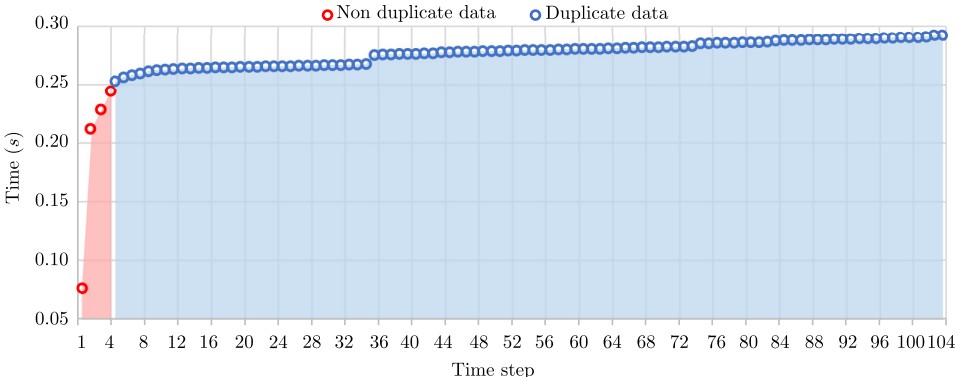

**Fig 6. Snapshot of sorting time change when there exists a mixture of 100 non-duplicate numbers and 100 duplicate numbers.**

## Data duplication

This section consists of two experiments to observe the gradient of sorting time when there are duplicates. The first experiment was conducted by using one synthetic data set. The data set consists of 100 non-duplicate numbers and 100 duplicate numbers. The size of working memory was set to 50 units. All non-duplicate numbers were equally divided into 4 chunks entering the sorting process one chunk at a time from time step 1 to time step 4. After that, a chunk of numbers with size as large as the available working memory was fed into the sorting process. The snapshot of each time step is shown in Fig 6. The first four steps show the run times of 100 non-duplicate data. Steps 5-104 illustrate the run time when only one duplicate number enters. The rate of run time change of non-duplicate data shown in the red area is greater than the rate of run time change of duplicate data shown in the blue area. After sorting all non-duplicate numbers, the run time change of duplicate numbers is almost linear and constant.

The second experiment concerning duplicate numbers was conducted by using real data sets containing duplicates. Three real-world data sets from kaggle.com, i.e. Delhi Weather Data [40], Artificial Lunar Landscape Dataset [41], and Continue Playing Game [42], were included in the experiment. The properties of attributes in these data sets are shown in Table 6. To relate the types of sub-sequences as defined previously, the distance average (Delta average) between two consecutively sorted input numbers is reported in Table 6. Four

**Table 6. Description of three real-world data sets.**

| Data sets | | Number of data | Delta average | Minimum | Maximum | Area |
|---|---|---|---|---|---|---|
| Delhi Weather Data | | | | | | |
| | Dew point | 100,369 | 0.98 | -24 | 75 | Weather |
| | Humidity | 100,233 | 7.18 | 4 | 243 | Weather |
| | Pressure | 99,843 | 0.74 | 929 | 1033 | Weather |
| | Temperature | 100,317 | 1.63 | 1 | 90 | Weather |
| Artificial Lunar Landscape Dataset | | | | | | |
| | Length | 18,867 | 50.74 | 21 | 720 | Image |
| | Height | 18,867 | 34.61 | 21 | 480 | Image |
| Continue Playing Game | | | | | | |
| | Hero_id | 17,485,730 | 23.93 | 21 | 720 | Game |

**Table 7. The time consumption, memory size for storing sorted data, and the ratio of data size to memory size for storing sorted data for three real-world data sets.**

| Data sets | Sorting time (sec) for different working memory sizes | | | | Final memory size used after sorting all data | Ratio of data size to memory size used |
|---|---|---|---|---|---|---|
| | 50% of data | 35% of data | 1% of data | 0.1% of data | | |
| Delhi | | | | | | |
| Dew point | 0.18 | 0.19 | 0.78 | 5.65 | 49 | 2,048 |
| Humidity | 0.28 | 0.34 | 3.31 | 15.23 | 103 | 973 |
| Pressure | 0.28 | 0.33 | 1.29 | 11.88 | 171 | 589 |
| Temperature | 0.24 | 0.27 | 1.43 | 8.12 | 52 | 1,929 |
| Artificial Lunar Landscape Dataset | | | | | | |
| Length | 0.48 | 0.58 | 5.72 | *Overflow* | 420 | 283 |
| Height | 0.45 | 0.52 | 3.73 | *Overflow* | 210 | 566 |
| Continue Playing Game | | | | | | |
| Hero_id | 1.52 | 1.46 | 3.37 | 14.95 | 65 | 269,011 |

different working memory sizes were set to 0.1%, 1%, 35%, and 50% of total input data size. The sorting time in seconds of each data set is reported in Table 7. When the working memory size was set to 0.1% of total input data, none of the data sets except the *Artificial Lunar Land-scape* data set could be sorted, as indicated by the word **Overflow**, because all spaces in the working memory were occupied by the compact groups and a set of single numbers after incoming data chunks. The final size of working memory used after sorting all data is also reported in column 6. Notice that the final size of working memory used is much smaller than the actual size of the sorted data set because the sorted data were compacted in the form of a compact set. For example, data set *Dew point* contains 100,369 numbers. After sorting, the size of final used memory is only 49 entries. The last column summarizes the ratio of data size to the final used memory size. In the case of *Dew point*, this ratio is $100,369/49 \approx 2,048$.

## Time complexity analysis

There are four main parts included in this analysis. Let $M$ be the working memory size, and $H$ be the remaining size of working memory after the sorting process for each incoming chunk of $h$ numbers. Generally, the remaining size is very small in comparison with the size of working memory and the total number of streaming data $n$, i.e., $H \ll M < n$, as shown in Fig 7.

The first part is the process of forming a *compact set* $Q^{(t)}$, where $t$ is the number of iterations. The time complexity of this part is $O(M)$. The second part is to update duplicate data. $h$ data must be analyzed to find their appropriate positions for insertion. The time complexity for updating duplicate data is $O(h \log M)$. The third part is to insert $h$ data into a compact set. The time complexity is $O(h \log M)$. If a datum $d_i$ is inserted into a *compact group*, $d_i$ and the compact group are transformed into a new compact group with time complexity of $O(1)$. Therefore, the time complexity of the third main part is $\max\{O(h \log M), O(h \log M) + O(1)\} =$

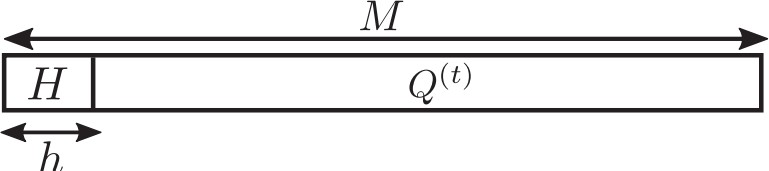

**Fig 7. Constraint on the limitation of working memory size, $H \ll M < n$.**

$O(h \log (M))$. The last part is to merge members in *compact set* $Q^{(t)}$. The merging algorithm is based on linear searching, and its time complexity is $O(M)$. In total, the time complexity of $n$ data is $O(M) + O(nh \log M) + O(nh \log M)) + O(nM)$. Because $h$ and $M$ are constant, the time complexity of *fast streaming data sort* is $O(n)$.

## Working space analysis of duplicate data

In *streaming data sort* [34], the lower bound of working memory size for correctly sorting non-duplicated $n$ numbers is $0.35n$. In the case of duplication, suppose that each number has only one duplicate, which indicates that there are $2n$ numbers to be sorted. Obviously, if *streaming data sort* is applied, then the size of working memory must be set to $0.35 \times 2n = 0.7n$. However, the size of working memory used by the proposed *fast streaming data sort* is less than $0.7n$. This is because all non-duplicate $n$ numbers can be sorted first by setting the size of working memory to $0.35n$. All duplicates do not need to be sorted, but only stored in a duplicate set $R^{(t)}$ as discussed in **Algorithm 1**. Each duplicate requires one entry in set $R^{(t)}$. Since there are $n$ duplicates, the size of set $R^{(t)}$ is only $n$ and the minimum size of working memory needed by *fast streaming data sort* is at least $0.35n + n$. This amount is equal to $\frac{0.35n+n}{2n} = 0.67$ or 67% of the data size. After the sorting process is finished, there must be only one compact group, which uses only two entries of memory space to represent the compact group, and $n$ entries for storing all $n$ duplicates.

## Discussion

Even though *fast streaming data sort* effectively sorts streaming data under the working memory limitation, there are two concerns about its limitation. First, the working memory size is 35% of the input data size. This means the proposed algorithm can reduce the working memory size up to 65%. On the other hand, the number of incoming data that can be handled by the proposed algorithm is up to $\frac{1}{0.35} = 2.86$ times the working memory size without upgrading the memory space. Second, the proposed algorithm was designed for integer data, but it can be applied to sort floating-point numbers by multiplying each floating number with $10^a$, for an integer constant $a$. The value of $a$ must be predetermined from the number of digits in decimal part.

## Conclusion

A new sort concept is presented in order to greatly hasten the process of *streaming data sort* with limited working memory size and to efficiently cope with duplicate data and data lifecycles without increasing the time complexity as reported in [34]. Instead of unfolding each compact group to insert new incoming numbers in their correct orders, the computational concepts of insert position, size of compact group, and detecting duplicates are developed to eliminate those unnecessary steps proposed in [34]. Several relevant theorems were established to support the proposed concepts of new *streaming data sort* under new constraints. The new sorting algorithm named *fast streaming data sort* can achieve the time complexity of $O(n)$, where $n$ is the total amount of input numbers, and the space complexity is equal to 35% of all incoming non-duplicate numbers, which is the possible lower bound. With duplicates, however, the space complexity can be at most 67% of all incoming non-duplicate and duplicate numbers.

The results were compared with several classic sorting algorithms and external merge sort on streaming data sort and analyzed with respect to large synthetic and real-world data sets. The sorting time of the proposed algorithm is 28.18 times less than streaming data sort. The

proposed algorithm is faster than bubble sort and insertion sort when the size of input data is greater than 10,000 and faster than quick sort and merge sort when the size of input data is greater than 1,000,000. Moreover, the proposed algorithm is 1,452 times faster than external merge sort. Therefore, the proposed algorithm can achieve a remarkably faster sorting speed in comparison with the compared algorithms, even though memory overflow may occur in those classic sorting algorithms.

Theoretically, most of existing sorting algorithms cannot be applied to streaming data when working memory size is less than the size of incoming data. Although some algorithms are designed to cope with this situation, the sorting result is obtained with approximate orders and values. However, *fast streaming data sort* can sort streaming data and generate the sorting result with the exact orders and values on the entire streaming sequence by using only 35% of data size as the size of available working memory. In the other hand, if the memory capacity is limited, *fast streaming data sort* is robust to overflow issue caused by increasing incoming data rather than other existing algorithms. In practical point of view, *fast streaming data sort* can be applied to several real-world applications as follows.

1. Reducing the searching time of enormous input data by using search algorithm on the sorting result in compact group form.

2. Finding order statistics, quartile, decile, and percentile of huge streaming data in social network applications on mobile phones, data cleaning, or data querying.

3. Examining duplicate data to reduce cost of storing data, to find error detection of systems, or to delete data in the data cleaning process.

Although *fast streaming data sort* can cope with the duplicate streaming data under the limitation of working memory less than the size of input data, there are two issues to be considered in the future work as follows.

1. Adding or improving new patterns of compact sequence to generate new compact group types that can be adopted to all different consecutive numbers.

2. Developing a compression procedure for compact groups with several data types, for instance, floating point number or text string, resulting in compatibility of data in many areas such as business, forecasting, logistics, and science.

## Author Contributions

**Conceptualization:** Suluk Chaikhan, Suphakant Phimoltares, Chidchanok Lursinsap.

**Data curation:** Suluk Chaikhan.

**Formal analysis:** Suluk Chaikhan, Suphakant Phimoltares, Chidchanok Lursinsap.

**Investigation:** Suluk Chaikhan, Suphakant Phimoltares, Chidchanok Lursinsap.

**Methodology:** Suluk Chaikhan, Chidchanok Lursinsap.

**Project administration:** Suphakant Phimoltares.

**Software:** Suluk Chaikhan.

**Supervision:** Suphakant Phimoltares, Chidchanok Lursinsap.

**Validation:** Suphakant Phimoltares, Chidchanok Lursinsap.

**Visualization:** Suluk Chaikhan.

**Writing – original draft:** Suluk Chaikhan.

**Writing – review & editing:** Suphakant Phimoltares, Chidchanok Lursinsap.

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
