## [Decision Letter · Decision Letter 0]

5 Jan 2022

PONE-D-21-39238Fast Continuous Streaming Sort in Big Streaming Data Environment under Fixed-Size Single StoragePLOS ONE

Dear Dr. Phimoltares,

Thank you for submitting your manuscript to PLOS ONE. After careful consideration, we feel that it has merit but does not fully meet PLOS ONE’s publication criteria as it currently stands. Therefore, we invite you to submit a revised version of the manuscript that addresses the points raised during the review process.

ACADEMIC EDITOR:Many recent papers are missingThe limitations and drawbacks of the method should be highlighted ==============================

We look forward to receiving your revised manuscript.

Kind regards,

Mohamed Hammad, Ph.D.

Academic Editor

PLOS ONE

Journal Requirements:

2. Please list all data sources used in your Data availability statement. Please also state whether or not sharing the Kaggle data with the manuscript complies with the terms and conditions of Kaggle.

"This work was supported by Development and Promotion of Science and Technology Talents Project (DPST) and Thailand Research Fund under grant number RTA6080013."

"Suluk Chaikhan received a grant from Development and Promotion of Science and Technology Talents Project (DPST).

Chidchanok Lursinsap received a grant from Thailand Research Fund under grant number RTA6080013."

Reviewers' comments:

Reviewer's Responses to Questions

**Comments to the Author**

1. Is the manuscript technically sound, and do the data support the conclusions?

Reviewer #1: Yes

Reviewer #2: Yes

2. Has the statistical analysis been performed appropriately and rigorously? 

Reviewer #1: Yes

Reviewer #2: No

3. Have the authors made all data underlying the findings in their manuscript fully available?

Reviewer #1: Yes

Reviewer #2: Yes

4. Is the manuscript presented in an intelligible fashion and written in standard English?

Reviewer #1: No

Reviewer #2: Yes

5. Review Comments to the Author

Reviewer #1: This paper proposes a new fast continuous streaming sorting in a big streaming data environment is proposed to cope with the constraint of storage overflow.

This is an interesting paper that deserve to be improved before any publication in journal:

• The abstract must summarize the performance evaluation results.

• The related work papers are not up to date in the domain where there are a lot of research papers are recently. In addition the review for most of the cited work is nebulous. The review failed to summarize critical details about each of the cited research study and its relevant pros and cons.

• Text is overlapped with image in some figures.

• Equations should be aligned, and their fonts must be matched in the manuscript. All the equations should be revised and realigned.

• The conclusions section should conclude that you have achieved from the study, contributions of the study to academics and practices, and recommendations of future works.

• The list of references should be reformatted and checked again to be matched with the journal requirement where a different styles and types are used. Please check some spells and typos.

• The authors should make their manuscript proofread by a native English speaker.

Reviewer #2: The manuscript, in its present form, contains several weaknesses. Adequate revisions to the following points should be undertaken to justify the recommendation for publication.

1. The contribution is not stated clearly.

2. The abstract must be rewritten, please delete unnecessary sentences.

3. It is recommended to re-examine and design the Figures.

4. A deep and detailed comparison with other methods is mandatory.

5. Please add the system specifications used for the evaluation as well as the programming language.

6. Specify the limitations and drawbacks of the proposed method.

7: Please add future work to the conclusion section and discuss it briefly.

6. PLOS authors have the option to publish the peer review history of their article (what does this mean?). If published, this will include your full peer review and any attached files.

Reviewer #1: No

Reviewer #2: No

---

## [Author Response · Author response to Decision Letter 0]

5 Mar 2022

We would like to resubmit the revised manuscript entitled “Fast Continuous Streaming Sort in Big Streaming Data Environment under Fixed-Size Single Storage” (PONE-D-21-39238) to be considered for publication as an original article in the PLOS ONE. We appreciate your concerns and have improved the manuscript according to the advice of the academic editor and two reviewers. We detail the revision in 'cover-letter' and 'response to reviewers' files. We hope you find our manuscript suitable for publication and look forward to hearing from you.

---

## [Decision Letter · Decision Letter 1]

18 Mar 2022

Fast continuous streaming sort in big streaming data environment under fixed-size single storage

PONE-D-21-39238R1

Dear Dr. Phimoltares,

We’re pleased to inform you that your manuscript has been judged scientifically suitable for publication and will be formally accepted for publication once it meets all outstanding technical requirements.

Kind regards,

Mohamed Hammad, Ph.D.

Academic Editor

PLOS ONE

Additional Editor Comments (optional):

Reviewers' comments:

Reviewer's Responses to Questions

**Comments to the Author**

1. If the authors have adequately addressed your comments raised in a previous round of review and you feel that this manuscript is now acceptable for publication, you may indicate that here to bypass the “Comments to the Author” section, enter your conflict of interest statement in the “Confidential to Editor” section, and submit your "Accept" recommendation.

Reviewer #1: All comments have been addressed

Reviewer #2: All comments have been addressed

2. Is the manuscript technically sound, and do the data support the conclusions?

Reviewer #1: Yes

Reviewer #2: Yes

3. Has the statistical analysis been performed appropriately and rigorously? 

Reviewer #1: Yes

Reviewer #2: Yes

4. Have the authors made all data underlying the findings in their manuscript fully available?

Reviewer #1: Yes

Reviewer #2: Yes

5. Is the manuscript presented in an intelligible fashion and written in standard English?

Reviewer #1: Yes

Reviewer #2: Yes

6. Review Comments to the Author

Reviewer #1: I have reviewed the author's response to concerns and feel that they have adequately responded to them. The changes in paper text also sufficiently address the concerns - in my view, the paper is ready for acceptance.

Reviewer #2: The paper titled (Fast continuous streaming sort in big streaming data environment under fixed-size single storage) accepted from my side.

7. PLOS authors have the option to publish the peer review history of their article (what does this mean?). If published, this will include your full peer review and any attached files.

Reviewer #1: No

Reviewer #2: No

---

## [Editor Report · Acceptance letter]

25 Mar 2022

PONE-D-21-39238R1 

Fast continuous streaming sort in big streaming data environment under fixed-size single storage 

Dear Dr. Phimoltares:

I'm pleased to inform you that your manuscript has been deemed suitable for publication in PLOS ONE. Congratulations! Your manuscript is now with our production department. 

Kind regards, 

on behalf of

Dr. Mohamed Hammad 

Academic Editor

PLOS ONE